# Functional recovery prediction during rehabilitation after rotator cuff tears by decision support system

Aušra Adomavičienė[1], Kristina Daunoravičienė[2]*, Girūta Kazakevičiūtė-Januškevičienė[3], Romualdas Baušys[3]

1 Faculty of Medicine, Department of Rehabilitation, Physical and Sports Medicine, Vilnius University, Vilnius, Lithuania, 2 Department of Biomechanical Engineering, Vilnius Gediminas technical University, Vilnius, Lithuania, 3 Faculty of Fundamental Sciences, Department of Graphical Systems, Vilnius Gediminas technical University, Vilnius, Lithuania

☯ These authors contributed equally to this work.
* kristina.daunoraviciene@vilniustech.lt

## Abstract

### Background

Today's rehabilitation decision-making still relies on conventional methods and different specific targeted rehabilitation protocols. Our study focuses on the decision support system for early rehabilitation after rotator cuff (RC) tears repair, where a multicriteria decision-making framework (MCDM) is applied for the prediction of successful functional recovery and selection of a rehabilitation protocol.

### Objective

To identify factors that affect recovery outcomes and to develop a decision support system methodology for predicting functional recovery outcomes at early rehabilitation after RC repair.

### Methods

Twelve rehabilitation experts were involved in the design, calibration, and evaluation of a rehabilitation protocol based on the proposed decision support system constructed using the MCDM framework. For the development of a decision support system, 20 patients after RC surgery undergoing outpatient rehabilitation were enrolled in a prospective cohort clinical trial.

### Results

The MCDM framework (SWARA method) sensitively assesses different criteria and determines the corresponding criteria weights that were similar to criteria weights assessed subjectively by rehabilitation experts. The assignment of patients into the classes, according to the heuristic evaluation method based on expert opinion and the standard qualitative

**Data Availability Statement:** All relevant data are within the manuscript and its Supporting Information files.

**Funding:** The author(s) received no specific funding for this work.

**Competing interests:** The authors have declared that no competing interests exist.

evaluation methods showed the validity of MCDM methods remain the best new alternative in predicting recovery during rehabilitation

## Conclusions

The results of this paper show that sustainable rehabilitation is an area that is quite suitable for the use of MCDM. The most of rehabilitation protocols are based on traditional methods and approaches, but the sensitive results showed the validity of MCDM methods and remains the best new alternative in prediction recovery protocols during rehabilitation.

## Introduction

Rotator cuff tears (RCTs) are a prevalent condition that often causes disability. They are also a common cause of shoulder pain, and can result in a frozen shoulder, weakness, dysfunction and limitation of daily activities (work and sports). The symptomatic illness affects between 4% and 32% of patients with RC tears. Successful management of recovery of RC tears depends on adequate rehabilitation, whether non-surgical or post-surgical. The primary objectives of postoperative rehabilitation after RC repair are to protect the healing process and prevent joint stiffness and muscle atrophy concurrently. However, there are not many evidence-based studies regarding postoperative rehabilitation protocols. While some authors advocate a traditional-conservative program [1], others argue that incorporating both protective and accelerated rehabilitation protocols, tailored to the individual case, can reduce the risk of postoperative stiffness without compromising the final result [2]. Several factors directly impact rehabilitation. First, these include medical factors related to tendon rupture, such as surgical approach, quality of the tendon, localization, and configuration of the rupture, and etiology of the rupture (degenerative or traumatic). Additionally, other influential factors encompass sociodemographic considerations, rehabilitation programs and measures [3]. Rehabilitation after RC repair starts with close communication between the patient, rehabilitation team, and surgeon, and must persist throughout the recovery process. The physical therapist must gather all relevant information to create an appropriate and successful rehabilitation protocol. These protocols can vary among providers and hinge on factors such as the selected progression time and therapeutic exercises. Whether option for a conservative or moderate rehabilitation protocol, the primary goals remain consistent: to maintain repair, alleviate tendon stress and pain, and facilitate the patient's return to the patient's previous daily activities.

Previous reviews [4,5] highlight the insufficient attention given to the evaluation of rehabilitation protocols and measures after RC repair. Typically, studies focus on assessing the efficiency, intensity and duration of conventional rehabilitation measures. Alternatively, some examine the pain syndrome, protocol to reduce shoulder stiffness, and techniques that aimed at enhancing functional recovery and reducing disability after rehabilitation, often analyzed separately. Factors contributing to successful or unsuccessful rehabilitation outcomes are typically analyzed including shoulder motor function (ROM, muscle strength, and endurance), shoulder mobility and stiffness, pain syndrome, functional problems, activities in daily living, and disability. Many known risk factors influencing functional recovery and rehabilitation efficiency often lead to postoperative stiffness, such as adhesive capsulitis, concomitant labral repair, single tendon rotator cuff repair, coexisting calcific tendinitis, pain syndrome and shoulder disability [6]. To mitigate these consequences, an early rehabilitation team should commence with a clear and detailed assessment of complex individual patient's functional and physical state, aided by prognostic decision-making assistance.

So, what constitutes best rehabilitation protocol?! How about one based on precise and competent decisions, directed purposefully and individually to the dysfunction and its causes? Decision-making is the process of making choices by identifying a decision, gathering information, and weighing alternative resolutions. Decision theories, based on mathematical, prognostic and probabilistic methods, have been proposed in various fields over the last century. The theory of multi-criteria decision-making (MCDM) was introduced in the second half of the 20th century. aiding the decision-makers in solving problems involving interacting criteria that require evaluation. MCDM stands as one of the most widely used decision methodologies across various fields, including energy and environment, business, economy, material selection, computer software selection, etc. [7,8]. Its application in biomedical engineering and healthcare is not a new approach and provides highly beneficial for medical professionals, patients, engineers and others in the field [9,10]. However, in recent years, an effective methodology using MCDM has not yet been widely applied in rehabilitation. Yet, in related medical fields such as oncology, MCDM is employed when choosing a treatment strategy (operative or conservative treatment), and drug treatment algorithms are based on MCDM [11].

Functional assessment in rehabilitation is conducted by a team, utilizing various methods for both assessment and protocol implementation including the most recent and effective main tools, methods, scales. However, until now, rehabilitation has seen only attempts to apply prognostic methods, and multifactorial analysis methods. These methods aim to optimally create a rehabilitation protocol and select measures after a detailed analysis of various factors that determine the success of recovery and effective rehab outcomes.

Today's decision-making in rehabilitation still relies on routine methods and various specific targeted rehabilitation protocols. Our study is focused on early rehabilitation after rotator cuff tears, where decision-making is carried out according to numerous rehabilitation protocols for the management of RC disease. These protocols are primarily based on clinical experience and expert opinion. Clinical doctors use MCDMs, including multi-rule decision-making, multi-objective decision-making, and multi-attribute decision-making, to analyze healthcare problems from multiple perspectives [9,11]. In practical healthcare cases, numerous critical parameters (criteria) can directly or indirectly influence the consequences of different decisions. Semi-structured and unstructured decision-making issues involve multiple criteria (or objectives) that may conflict with each other [11]. Therefore, the utilization of MCDM becomes a promising solution for practical problems. The importance of multi-criteria methods lies in the ability to assess the weights of criteria using appropriate methods (such as The Step-wise Weight Assessment Ratio—SWARA) [7], allowing the evaluation of the superiority of the criteria in relation to each other. MCDM can be defined as decision support/assistance in decision-making by assessing the patient's health state, functional and physical capabilities and monitoring the effectiveness and recovery of different functions at the beginning of the rehabilitation program after RC repair.

Therefore, in this paper, we developed and tested an MCDM-based evaluation methodology for determining the optimal recovery approach in rehabilitation for RC pathology.

We hypothesized that MCDM can assist a rehabilitation specialist in developing a detailed rehabilitation protocol for each patient after RC tears surgery, taking into account the patient's individual physical, functional and emotional state, as well as postoperative risk and needs. In other words, the goal is to select criteria from numerous factors and, using multifactorial methods, gradually refine those that will aid in deciding on the most optimal rehabilitation protocol. Therefore, the objective of the study is to identify factors that affect recovery outcomes and to develop a decision-support methodology for predicting functional recovery outcomes in early rehabilitation after RC repair.

The second chapter is dedicated to the analysis of related state-of-the-art scientific works, encompassing and examination of the methodologies used for evaluating successful rehabilitation (both classical and MCDM approaches). In the third chapter, we detail the methodology applied in our work focusing on a particular selected case. Finally, the fourth and fifth chapters present the results obtained and provide discussion statements.

## Analysis of works related to the topic

Rehabilitation protocols after RC repair are designed to protect recovery in the immediate postoperative period while addressing postoperative stiffness and muscle atrophy and pain and functional issues. Recent studies have yielded mixed results for postoperative treatment modalities and rehabilitation protocols. Conservative and accelerated rehabilitation protocols did not exhibit any significant difference, and early passive range of motion (ROM) after RC repair demonstrated reduced postoperative stiffness and improved functionality. Researchers reported the effectiveness of exercise on functional recovery and provided comparisons of various postoperative rehabilitation methods, including a land-based program with or without aquatic therapy, individualized physical therapy plus exercise at home, an inpatient outpatient rehabilitation program, progressive weight-bearing rehabilitation, home exercise instruction, etc. All the rehabilitation protocols and treatments presented in Table 1 are primarily associated with the main symptoms accompanying patients after RC repair and how they are alleviated.

The functional recovery after RC repair and rehabilitation outcomes are influenced not only by various factors associated with RC pathology but also by complex and individually assigned rehabilitation measures. This includes considerations such as intensity and duration, whether it aligns with the patient's individual physical and psycho-emotional characteristics, tolerance of physical load, performance of independent tasks, and engagement in daily physical activities.

**Table 1. Various treatment and rehabilitation protocols after RC repair.**

| No. | Rehabilitation protocol/treatment methods | Targeted outcome | Author (Publication Year) |
|-----|------------------------------------------|------------------|---------------------------|
| 1. | Early rehabilitation protocols | Repair integrity, shoulder functional recovery, pain reduction/ improved quality of life | Berton, A. et al. (2021) [4]; Fahy, Kathryn et al. (2022) [5]; Jennifer C. Seida et al. (2010) [6]; Roe Y. et al. (2013) [12] |
| 2. | A land-based program with or without aquatic therapy | Reduced pain and restored shoulder function | Byung-Su Kim et al.(2021) [13]; Wang, H., Hu, F., Lyu, X. *et al.* (2022) [14]; |
| 3. | An inpatient rehabilitation program | Reduced pain and restored shoulder function | Altintas, Burak et al. (2020) [15]; Singh Jagdev, Balraj et al. (2022) [16]; |
| 4. | An outpatient rehabilitation program/ home exercise instruction/ early return to daily life activities | Reduced pain and restored shoulder function | Jennifer C. Seida et. al. (2010) [6]; Altintas, Burak et al. (2020) [15]; Daghiani, Maryam et al. (2022) [17]; Byung-Su Kim et al.(2021) [13] |
| 5. | Progressive weight-bearing rehabilitation | Significant reduction in pain | Singh Jagdev, Balraj et al. (2022) [16]; Yi, D et al. (2021) [18]; Brochin, Robert L et al. (2020) [19]; Altintas, Burak et al. (2020) [15]; Kennedy, Patrick et al. (2019) [20] |
| 6. | Postoperative nonsteroidal anti-inflammatory drugs during the first week after surgery | Pain control and reduction in 2–3 months after RC tears repair | Daghiani, Maryam et al. (2022) [17]; Audigé, Laurent et al. Audigé, Laurent et al. (2021) [21] |

**Table 2. Functional recovery influencing factors in order of importance.**

| No | Influencing factor | Evaluation methods and instruments | Rehabilitation protocol/treatment methods | Author (Publication Year) |
|---|---|---|---|---|
| 1. | Shoulder pain and dysfunction | Visual Analog Scale for Pain (VAS) | Anti-inflammatory drugs, comfortable hand positions, Various exercises, Massages and Physiotherapeutic measures | Brochin, Robert L et al. (2020) [19]; B.G. Lee et al. (2012) [22]; Yi, D et al. (2021) [18]; Adomavičienė et all. (2021) [23] |
| 2. | Shoulder functional parameters | Patient-reported questionnaire, ROMs measurement, Apley scratch test, manual muscle testing scale, spring scale, dynamometry, DASH | | Sun, Zhengyu et al. (2018) [24]; Wong, W.K. (2020) [25]. Jain, Nitin B et al. (2018) [26] Byung-Su Kim et al. (2021) [13] S. Hajivandi (2021) [27] |
| 3. | Psychoemotional state: anxiety and depression; sleep restriction | Psychological distress and patient-reported subjective well-being | Pain relief methods, inpatient rehabilitation program, inclusion in daily life activities | Park, Joo Hyun et al (2021) [28]; O.Nikolaidou (2017) [3] |
| 4. | Age | Birth data | Inclusion and participation in daily life activities | K.Saito et all. (2021) [2]; Z.Wani et all (2016) [29]; Wu XL et al. (2012) [30]; Audigé, Laurent et al. (2021) [21]; Oh, Luke S et al. (2007) [31] |
| 5. | Duration of symptoms | Pain syndrome, shoulder stiffness and disability | Exercises and training programs, Massages and Physiotherapeutic measures | Jennifer C. Seida et al. (2010) [6]; Oh, Luke S et al. (2007) [31]. |

Different factors influencing functional recovery are evaluated in rehabilitation, and, while all are important, their impact on functional status depends on many circumstances. Scientists and medical professionals have identified the most crucial, most influential factors after RC tears, recommending, close monitoring of these factors (see Table 2).

RC tear is a major cause of shoulder pain and dysfunction typically classified as nociceptive pain [26]. Most scientists focus on individual factors influencing pain, impacting various aspects of recovery. These factors include defining the duration of pain in days, weeks and months, accurately assessing the nature of pain using qualifying adjectives such as intensive, intermittent, sharp, burning and etc., and identifying the causes of the pain intensification (during movements, specific positions or exercises) or the presence of dynamic pain (during movements) and factors contributing to pain reduction (during movements, specific positions, exercises, or after therapies) [19,32]. In contrast, a lot of studies emphasize the potential importance of accounting for concomitant rehabilitation measures as predictors of early postoperative pain intensity outcomes and successful pain control protocol. This consideration is vital due to the reduced motor function of the shoulder, arm and hand and increased disability observed as consequences of the long-term effect of pain, immobility and muscle stiffness, shoulder instability and limitation of daily life activities [22,33]. Many researchers assert a

direct relationship between the patient's psycho-emotional state, expectations, the presence of anxiety and the results of rehabilitation [18,34]. Sleep disorders are commonly observed in patients after RC repair, there is an increase in the quality of sleep with a parallel improvement in shoulder functions. Consistent improvements in sleep quality are noted within the 6 months after RC repair and are significantly related to physical recovery and decreased anxiety [34]. Age is one of the dominant factors in RC tear and recovery mechanisms [18,29]. Generally, RC tears are more prevalent in the elderly than in younger patients, being chronic or acute and chronic and usually secondary to tendon degeneration. The literature shows an increasing prevalence of RC tears in the elderly, with a mean age of 60.2 years (range, 23–81 years) [25]. Some studies suggest that a duration of symptoms exceeding 1 year is one of the main negative prognostic factors for successful recovery [24]. A substantial percentage of patients who experienced symptoms for less than 3 months reported lasting improvement [16,29].

Increasingly, medical professionals acknowledge that the functional assessment of the patient and data collection have become routine aspects of their work, emphasizing the need for an additional auxiliary instrument [13–15]. An algorithm, as such an additional tool, would assist in indicating the main guidelines, and criteria for the creating an individual patient's rehabilitation program.

## Methodology

### Framework of the methodology

This section illustrates the framework of the methodology for identifying criteria, selecting a rehabilitation protocol after RC repair, and classifying / grouping patients into appropriate classes based on the success of rehabilitation. To address the objective of our study, i.e. making the optimal decision, the main steps for the multi-criteria decision-making methodology were included: (1) identifying the problem using routine functional assessment methods in rehabilitation, discussing factors influencing recovery, and determining their importance through expert discussion, (2) defining criteria and sub-criteria and evaluating their weights (3) processing data using normalization and MCDM methods for indicators aggregation, and (4) reporting results Fig 1 represents the methodological steps of our study.

### Research team

Twelve rehabilitation specialists participated as experts in the experimental study comprising 2 Physical and Rehabilitation (PRM) doctors with over15 years of rehabilitation experience, 6 physiotherapists and 4 occupational therapists, all with 15–10 years of experience in rehabilitation with RC pathology. All experts hold a scientific degree (PhD) and are actively engaged in scientific research.

### Participants

The problem definition begins with patient selection based on inclusion and exclusion criteria. Inclusion criteria are as follows (1) age $\geq$ 40 years, (2) presence of aa full-thickness tear of the rotator cuff muscles, (3) 6–7 weeks post-surgery, (4) first-time surgery, and (5) voluntary agreement participate in the study. Exclusion criterion include: (1) RC tear retraction and repeated shoulder surgery, (2) co-morbidities affecting shoulder function; (3) more than 8 weeks post-surgery.

The prospective cohort clinical study included 20 patients after RC surgery, undergoing outpatient rehabilitation at Vilnius University Hospital's Rehabilitation Center. Commencing 6–7 weeks after surgery, participants engaging in a 2-week conventional outpatient rehabilitation program. 4 patients (16.6%) did not complete the rehabilitation program due to

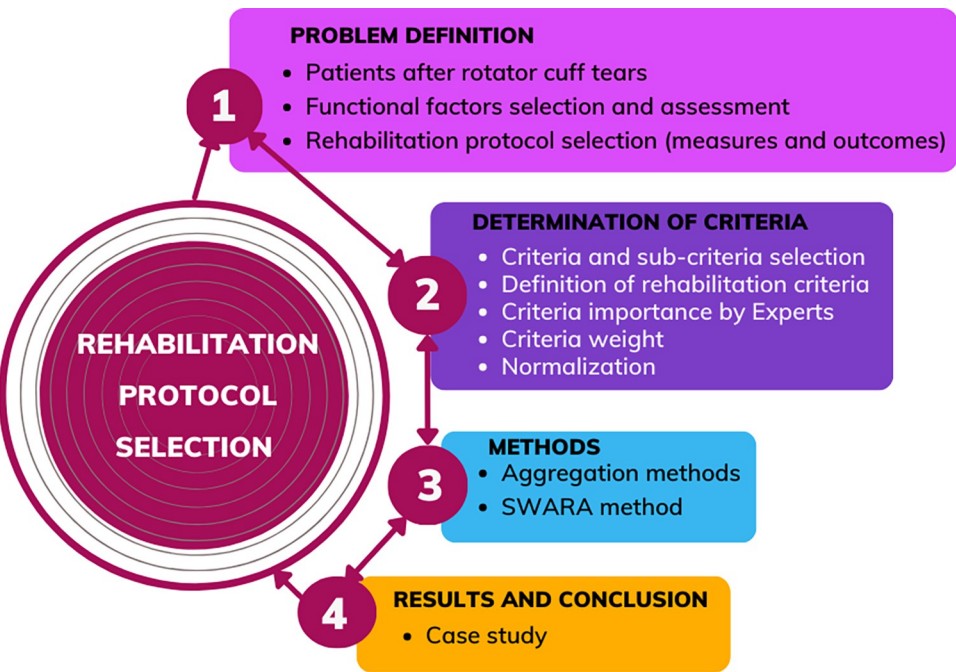

**Fig 1. Methodical steps of the workflow.**

worsening health including pain, high blood pressure, heart rate problems and intolerance of physical load. The assessment data were collected, scored, and entered into a database by a study coordinator. Patient consent to participate in the study was obtained after informing them about all the details. The data collection followed to Ethical Protocol provided by the Vilnius Regional Biomedical Research Ethics Committee (No.2021/5-1349822), and informed written consent was obtained. The recruitment period for this study was from 21.06.2021 to 20.12.2022.

## Different functional factors selection

In this section, we present a comprehensive theoretical model outlining the selection of various factors related to functional and physical states for a rehabilitation protocol. These factors are not individual health condition restrictions but rather an integrated choice suitable for subsequent protocol creation for each patient. Our proposed protocol constitutes a multi-criteria analysis system structured according to two main levels: the patient's physical-functional state and psycho-emotional state (Fig 2). It is important to note that the usual rehabilitation program (duration and intensity) and the measures applied to patients after RC repair adhere to the requirements of the national rehabilitation medical regulation.

The study commenced after a thorough evaluation of practically all possible selection criteria of patients after RC repair, considering their functional and psycho-emotional condition, as well as factors related to RC repair and socio-demographic characteristics. The choice and importance of criteria were determined by 10 independent qualified experts specializing in RC pathology rehabilitation. Their decisions were informed by previous works' reports and their own practical experience. During the deliberations, the assessment of factors such as gender, socio-demographic state (education, employment and residence), dominated/not dominated arm and sports or physical recreation activities were excluded because they were deemed to have a significant impact on rehabilitation outcomes. It was unanimously agreed that the most

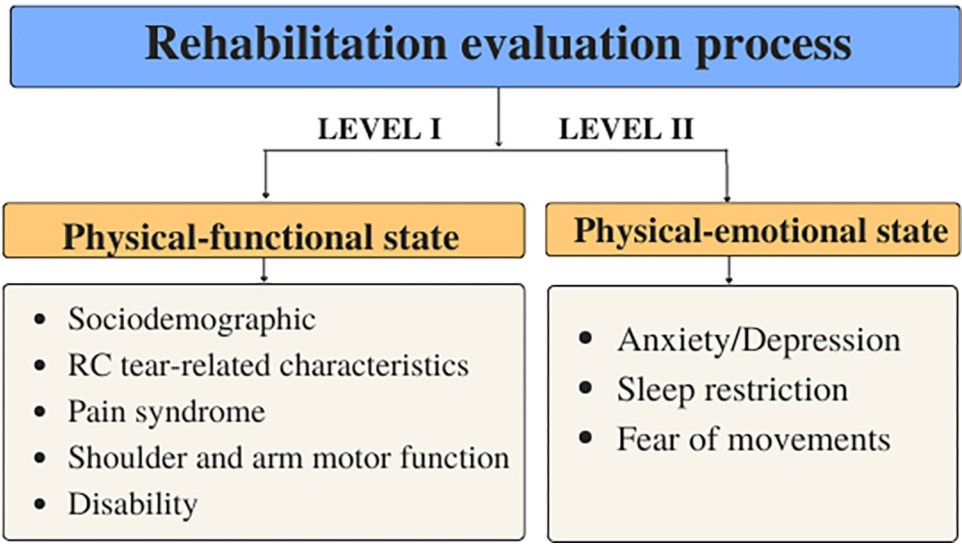

**Fig 2. Structure of proposed multicriteria system.**

important criterion of successful rehabilitation and functional recovery is the Pain syndrome (Intensity and duration of pain, nature of pain, causes of the intensification of pain, Intensity, nature and duration of dynamic pain (c1). The second important criterion is Shoulder motor function and disability, which includes Shoulder range of motion (ROM), muscle strength, hand grip strength, dexterity and functioning in daily life activities (c2). Most studies confirm that psycho-emotional factors such as anxiety, depression (c3), sleep restriction (c7) and fear of movements (c5) significantly influence the level of disability and pain experienced by patients before surgery. However, there is a notable improvement in postoperative pain and shoulder functional level, even in the presence of significant psychosocial impairment. While sociodemographic and injury-related characteristics (c4) should be given the highest priority. These factors and characteristics including age, causes of RC pathology/injury, affected muscle, time since surgery/rehabilitation and symptoms duration directly determine the treatment strategy (surgery or conservative treatment) and rehabilitation outcomes. Physical recovery and psycho-emotional well-being of patients after RC repair' are closely related to the presence of sleep restrictions (c6), which are also considered significant influencers for healing and functional recovery. The last group of criteria includes Healthcare organization and routine after RC repair—Rehabilitation measures (c7), Exercising intensity and duration (c8), Home activities in daily life (c10) and Medicines (c9). These criteria characteristics can aid clinical staff in selecting a recovery protocol and achieving successful rehabilitation outcomes. Additionally, they can help identify patients prone to re-tear early after RCT repair, allowing for the development targeted prevention and treatment strategies for modifiable risk factors.

The criteria were sorted based on the experts' qualifications and experience related to the rehabilitation after RC tears. According to the experts, three groups of criteria were singled out: (I) Physical and functional condition, (II) Psycho-emotional condition, and (III) Healthcare organization and routine, and were ranked according to importance (see Table 3).

Additionally, each group of criteria was separated into sub-criteria, and they were ranked according to importance, expressed as a percentage and numbers. The experts ranked the factors in order of importance ensuring consistency in their views. Based on the criteria and the ranking of sub-criteria by importance presented in Table 4, a pairwise assessment of the relative importance of the criteria was performed (Table 4).

**Table 3. Criteria (c1 – c10) sorted by relevance.**

| No | Min/max | Criteria title | Descriptive parameters / Problem considered | Author (Publication Year) |
|---|---|---|---|---|
| c1 | Min | Pain syndrome | Intensity and duration of pain, Nature of pain; Intensity, nature and duration of dynamic pain | Eivind Inderhaug (2018) [35]; Altintas, Burak et al. (2020) [15]; Singh Jagdev, Balraj et al. (2022) [16]; Kennedy, Patrick et al. (2019) [20]; J.Tangtiphaiboontana (2021) [32]; Yi. D (2021) [18]; Audigé, Laurent et al. (2021) [21]; Brochin, Robert L et al. (2020) [19]; |
| c2 | Max | Shoulder motor function and disability | Shoulder ROM, muscle strength, dexterity of movements, Functioning and disability in daily activities | Kennedy, Patrick et al. (2019) [20] Brochin, Robert L et al. (2020) [19] Audigé, Laurent et al. (2021) [21] Jain, Nitin B et al. (2018) [26] Byung-Su Kim et al. (2021) [13] Shaahin Hajivandi (2021) [27] |
| **c3** | Min | Anxiety / Depression | Presence of anxiety / depression symptoms | Park, Joo Hyun et al. (2021) [28] Wong, W.K (2020) [25] A.Adomavičienė (2021) [23] Kennedy, Patrick et al. (2019) [20] (Yngve Roe, 2013) [12] |
| c4 | Min | Sociodemographic and with injury related characteristics | Age, causes of injury, time since surgery intervention, affected muscles | Kennedy, Patrick et al. (2019) [20] Wang, H., Hu, F., Lyu, X. *et al.* (2022) [14] Z.Wani (2016) [29] (Yngve Roe, 2013) [12] |
| c5 | Min | Fear of movements | Fear of shoulder movements | Wang, H., Hu, F., Lyu, X. *et al.* (2022) [14] Eivind Inderhaug (2018) [35] Kennedy, Patrick et al. (2019) [20] |
| c6 | Min | Sleep restriction | Presence of sleep restriction | Kyle N. Kunze (2020) [34] Kennedy, Patrick et al. (2019) [20] (Yngve Roe, 2013) [12] |
| c7 | Max | Rehabilitation program measures | Conventional complex rehabilitation program | Daghiani, Maryam et al. (2022) [17] Audigé, Laurent et al. (2021) [21] Brochin, Robert L et al. (2020) [19] |
| c8 | Max | Rehab program intensity and duration | 2–2.5 weeks | Brochin, Robert L et al. (2020) [19] Jain, Nitin B et al. (2018) [26] Byung-Su Kim et al.(2021) [13] Wang, H., Hu, F., Lyu, X. *et al.* (2022) [14] |
| c9 | Min | Medicines | Use of medication (analgesics, anti-inflammatory drugs) | Kennedy, Patrick et al. (2019) [20] Daghiani, Maryam et al. (2022) [17] Audigé, Laurent et al. (2021) [21] |
| c10 | Max | Home activities in daily life and exercising intensity | Intensity of exercising and ADL activity at home | Daghiani, Maryam et al. (2022) [17] Byung-Su Kim (2021) [13] Wang, H., Hu, F., Lyu, X. *et al.* 2022) [14] Eivind Inderhaug (2018) [35] |

**Table 4. Criteria and sub-criteria ranking according to importance.**

| | Criteria and sub-criteria | | Percent | Type |
|---|---|---|---|---|
| | **Physical and functional state** | | **50%** | |
| c1 | **Pain syndrome** | **Ranking by numbers** | **40%** | |
| c1.1 | Intensity of the pain | 1 point–mild pain; 2 points-moderate pain; 3 points-severe; 4 points-most severe pain | 35% | Min |
| c1.2 | Duration of the pain | 1 point–<7 days, 2 points–<4 weeks, 3 points–<3 month, 4 points–<6 month, 5 points > 6 month | 20% | Min |
| c1.3 | Nature of the pain | 1 point–intermittent pain, 2 points–dynamic, 3 points–sharp, 4 points–intensive | 18% | Max |
| c1.4 | Dynamic pain | 1 points–< 90 points, 2 points–> 90 points | 12% | Max |
| c1.5 | Pain increasing factors | 1 point—exercising; 2 points—movements; 3 points - position; 4 points—during rest | 10% | Max |
| c1.6 | Pain decreasing factors | 1 point–position, 2 points—movements, 3 points—during rest, 4 points—exercising, | 5% | Min |
| c2 | **Shoulder motor function and disability** | | **35%** | |
| c2.1 | Shoulder ROM | 1 point—up to (flexion <90, abduction < 90); 2 points—over (flexion >95, abduction > 95) | 40% | Max |
| c2.2 | Shoulder muscle strength | 1 point– 0, 2 points– 1, 3 points– 3, 4 points—4 / 5 | 30% | Max |
| c2.3 | Shoulder functioning and disability | 1 point—0–5, 2 points—6–10 | 15% | Min |
| c2.4 | Shoulder, arm and hand functioning in ADL | 1 point <54, 2 points—55–68, 3 points—69–85, 4 points—86–100 | 10% | Min |
| c2.5 | Dexterity of movements | 1 point—<69 blocks, 2 points—70–79 blocks, 3 points—>80 blocks | 5% | Max |
| c4 | **Sociodemographic and with injury related characteristics** | | **25%** | |
| c4.1 | Affected muscles | 1 point–single affected muscle, 2 points–two affected muscle, 3 points–> two affected muscles | 40% | Min |
| c4.2 | Time since surgery | 1 point - <6 days, 2 points - <2 weeks, 3 points—<4 weeks, 4 points - >5 weeks | 30% | Min |
| c4.3 | Age | 1 point– 55–60; 2 points– 61–75; 3 points > 76–85 | 15% | Min |
| c4.4 | Causes of injury | 1 point - Trauma 2 points - Degenerative | 10% | Min |
| c4.5 | Affected arm | 1 point—Dominated, 2 points—Not dominated | 5% | Min |
| | **II. Psychoemocional status** | | **30%** | |
| c3 | Anxiety / Depression | 1 point 0–7, 2 points—8–10, 3 points 11–14, 4 points 15–21 | 35% | Min |
| c5 | Fear of movements | 1 point—1–36; 2 points—37–68 | 20% | Min |
| c6 | Sleep restriction | 1 point—normal sleep, 2 points -sleep restriction first week after surgery, 3 points -sleep restriction during rehabilitation, 4 points -sleep restriction after activities | 45% | Min |
| III. | **IV. Healthcare organization and routine** | | **20%** | |
| c7 | Rehabilitation measures (Conventional rehabilitation program) | | 40% | Max |
| c8 | Program intensity and duration (2–2.5 weeks) | | 20% | Max |
| c9 | Medicines | | 25% | Min |
| c10 | Home activities and exercising intensity | | 15% | Max |

All information concerning the importance of decision makers and individual expert evaluations regarding the rating of the criteria and sub-criteria via attributes and attribute weights usually are expressed in linguistics terms. The set of linguistics terms used to rate the importance of each criterion/sub-criteria for the decision-makers follows standard protocols used during rehabilitation. The ranking by numbers of criteria/ sub-criteria linguistics terms is presented in Results section (see Table 4, Column ranking by points).

## Follow-up routine

The conventional rehabilitation program following RC repair consists of measures such as physical therapy, occupational therapy, massage, physiotherapy, psychologist, social worker and patient education. The duration of rehabilitation program is 2–2.5 weeks with an intensity of 5 times per week and 3.5–4 hours per day following an individually prepared rehabilitation protocol with rest breaks between sessions. At the end of rehabilitation, patients are provided

with a home program containing individualized home physical activity recommendation including individually designed physical load in terms of intensity and duration.

The evaluation of the Physical and functional state of the patient was performed at the beginning (I assessment) and the end (II assessment) of outpatient rehabilitation. The assessment measures were the following: (1) Patients' sociodemographic characteristics and information related to RC pathology, rehabilitation measures, medicines and exercise intensity at home were recorded from the patient's medical history. (2) Pain syndrome—intensity, severity, and nature of pain were evaluated by VAS scale [18,26] and McGill pain questionnaire [16,25]. (3) Shoulder motor function was evaluated using manual muscle strength testing (MMT) [13,15,16], shoulder active ROM [14,20,27], and dexterity of movements by Box&Block test [36]. (4) Functioning and disability of the Arm, Shoulder and Hand by DASH [17,20,21] and SPADI [28]; (5) Psycho-emotional state–anxiety and depression evaluated by HAD scale [19,21,23,25,28,35], Sleep restrictions [25,34] and Fear of movements by TAMPA Scale of Kinesiophobia [14,30,31].

After ranking, the classification of patients into 5 classes was conducted using two methods: a heuristic evaluation method based on expert opinion and a classical evaluation method using the K-Means clustering algorithm. Following the calibration of the comparison of the classification into classes by the two methods, the exact limits of each class are determined, and the differences between the class distributions were evaluated. The heuristic method of grouping patients into classes based on expert opinion on patient recovery success and rehabilitation outcomes, was based on the methodology of the International Classification of Functioning, Disability and Health (ICF). This method utilizes a qualification system to assess the impairment of body functions, activity, limitations and participation restrictions in patients, those with shoulder disorders [12,37]. The final grouping of patients into classes according to rehabilitation outcomes is described as follows:

I class—Complete impairment: a problem that is present more than 95% of the time, with an intensity, that totally disrupts the person's daily life every day over the last 30 days;

II class—Severe impairment: a problem that is present more than 50% of the time, with an intensity that partially disrupts the person's daily life over the last 30 days;

III class—Moderate impairment: a problem that is present less than 50% of the time, with an intensity, that is interferes in the person's daily life over the last 30 days;

IV class—Mild impairment: a problem that is present less than 25% of the time, with an intensity a person can tolerate and which rarely happens over the last 30 days;

V class—No impairment—the person has no problem.

The second method of grouping patients into classes was performed using a standard qualitative assessment, applying one of the MCDM methods—the SWARA methodology to determine the weights of the criteria.

## MCDM methods

MCDM methods are generally used for health care decision-making problem: (1) nature of the decision/goal (sorting, ranking, and choice); (2) criteria type (qualitative/quantitative); (3) criteria structure (flat/hierarchical); (4) weight of criteria (subjective/objective); (5) alternatives (incremental/non-incremental) [7,11,38]. MCDM methods mainly comprise three parts: planning and design, evaluation and selection, and data processing. The data processing focuses on the analysis and identification of rehabilitation management problems and data features to address practical cases. The application provides a ranking result based on the selected criteria, their corresponding values, and assigned weights.

MCDM can be used in rehabilitation in a variety of ways. For instance, it can aid in prioritizing treatments based on the patient's individual needs and goals [9,10]. Additionally, it can

assess the effectiveness of different treatment options by comparing their outcomes against a set of predetermined criteria. One common MCDM approach used in rehabilitation is the Analytic Hierarchy Process (AHP) [39,40]. AHP involves breaking down a decision into a hierarchy of criteria and sub-criteria followed by assigning weights to each criterion based on its relative importance. These weights are determined through a pairwise comparison process, where the decision-maker compares each criterion to every other criterion and assigns a score based on their relative importance. These scores are then used to calculate the overall weight of each criterion in the decision. Another MCDM approach applicable in rehabilitation is the Technique for Order of Preference by Similarity to the Ideal Solution (TOPSIS). TOPSIS involves ranking different treatment options based on their similarity to an ideal solution [11,41]. The ideal solution is determined based on a set of predetermined criteria, and the treatment options are then ranked based on their distance from this ideal solution.

**SWARA method for criteria weighting.** When applying the MCDM framework to address real-life rehabilitation problems, it is crucial initially assess the different criteria and determine the corresponding criteria weights. The weighting of criteria importance is a key element in MCDM applications [38]. Various methods of criteria evaluation have been employed solving complex MCDM problems, such as AHP, SWARA, ANP and others [39]. In the proposed methodology, the SWARA (Step-wise Weight Assessment Ratio) technique [40] is utilized to define the main criteria weights, engaging experts that perform criteria ranking by importance in the criteria weighting process. This method is deemed superior to the AHP method due to a lower number of pairwise comparisons [41] and can be used successfully instead of ANP, FARE, and AHP methods [42].

The SWARA technique for the determination of the criteria weights can be described through the following steps:

1. Definition of the criteria set in rehabilitation (Table 3).

2. Arrangement of the criteria by their importance by experts (with the most important criteria in the first position, and in the least important in the last) (Table 4).

3. Evaluation of the comparative importance using the average value of the criterion importance $s_j$, starting from the second criterion. It involves determining how much criterion $s_j$ is more important than the criterion $s_{j+1}$ and do so for each following criterion (Table 5).

**Table 5. Evaluation of relative importance in pairs of criteria (in range from 0 to 1).**

| Experts | Pairwise evaluation of criteria relative importance | | | | | | | | |
|---|---|---|---|---|---|---|---|---|---|
| | $c_{1\leftrightarrow2}$ | $c_{2\leftrightarrow3}$ | $c_{3\leftrightarrow4}$ | $c_{4\leftrightarrow5}$ | $c_{5\leftrightarrow6}$ | $c_{6\leftrightarrow7}$ | $c_{7\leftrightarrow8}$ | $c_{8\leftrightarrow9}$ | $c_{9\leftrightarrow10}$ |
| 1 | 0.55 | 0.35 | 0.25 | 0.88 | 0.55 | 0.85 | 0.50 | 0.05 | 0.99 |
| 2 | 0.65 | 0.60 | 0.18 | 0.71 | 0.20 | 0.99 | 0.43 | 0.15 | 0.59 |
| 3 | 0.40 | 0.55 | 0.65 | 0.69 | 0.15 | 0.45 | 0.15 | 0.25 | 0.60 |
| 4 | 0.55 | 0.50 | 0.30 | 0.55 | 0.45 | 0.52 | 0.25 | 0.35 | 0.42 |
| 5 | 0.70 | 0.70 | 0.45 | 0.70 | 0.50 | 0.70 | 0.60 | 0.49 | 0.80 |
| 6 | 0.45 | 0.65 | 0.10 | 0.58 | 0.05 | 0.90 | 0.45 | 0.10 | 0.79 |
| 7 | 0.50 | 0.62 | 0.30 | 0.66 | 0.35 | 0.96 | 0.35 | 0.05 | 0.66 |
| 8 | 0.40 | 0.40 | 0.05 | 0.80 | 0.10 | 0.70 | 0.65 | 0.25 | 0.82 |
| 9 | 0.65 | 0.55 | 0.10 | 0.99 | 0.05 | 0.69 | 0.80 | 0.50 | 0.60 |
| 10 | 0.65 | 0.60 | 0.05 | 0.89 | 0.20 | 0.85 | 0.15 | 0.45 | 0.70 |
| 11 | 0.45 | 0.25 | 0.42 | 0.77 | 0.15 | 0.80 | 0.45 | 0.03 | 0.55 |
| 12 | 0.55 | 0.30 | 0.05 | 0.15 | 0.05 | 0.70 | 0.55 | 0.55 | 0.75 |
| Average values | 0.5417 | 0.5058 | 0.2417 | 0.6975 | 0.2333 | 0.7592 | 0.4442 | 0.2683 | 0.6900 |

**Table 6. Criteria and sub-criteria weighting by SWARA method.**

| Criteria | Average values of comparative importance criteria, $S_{j\leftrightarrow j+1}$ | Coefficients of comparative importance criteria, $k_j$ | Recalculated (intermediate) criteria weights, $q_j$ | Final Criteria weights, $w_j$ | Final Sub-Criteria weights, $w_j$ |
|---|---|---|---|---|---|
| c1 0.2 | – | 1.000 | 1.000 | 0.3292 | c1.1 0.11522 c1.2 0.0658 c1.3 0.0593 c1.4 0.0395 c1.5 0.0329 c1.6 0.0165 |
| c2 0.175 | 0.5417 | 1.5417 | 0.6486 | 0.2135 | c2.1 0.0854 c2.2 0.0641 c2.3 0.0320 c2.4 0.0214 c2.5 0.0107 |
| c3 0.135 | 0.5058 | 1.5058 | 0.4308 | 0.1418 | - |
| c4 0.125 | 0.2417 | 1.2417 | 0.3469 | 0.1142 | c4.1 0.0457 c4.2 0.0343 c4.3 0.0171 c4.4 0.0114 c4.5 0.0057 |
| c5 0.105 | 0.6975 | 1.6975 | 0.2044 | 0.0673 | - |
| c6 0.06 | 0.7592 | 1.7592 | 0.0942 | 0.031 | - |
| c7 0.08 | 0.2333 | 1.2333 | 0.1657 | 0.0545 | - |
| c8 0.05 | 0.4442 | 1.4442 | 0.0652 | 0.0215 | - |
| c9 0.04 | 0.2683 | 1.2683 | 0.0514 | 0.0169 | - |
| c10 0.03 | 0.6900 | 1.6900 | 0.0304 | 0.0100 | - |
| | | | **3.0376** | | |

4. Calculation of the comparative importance coefficient, as follows (Table 6, 3rd column):

$$k_j = s_j + 1$$

5. Determination of the recalculated weight is as follows (Table 7, 4th column):

$$q_j = \frac{q_{j-1}}{k_j}$$

6. The calculation of the final weights of criteria (Tables 8, 5th column):

$$w_j = \frac{q_j}{\sum_{j=1}^{n} q_j},$$

where $n$ is number of the criterion.

Criteria weights were assessed subjectively by expert. They were arranged in order of importance from the most to the least important. Subsequently, these experts compared adjacent criteria pairs. Values of pairwise comparison were presented in Table 5. Table 6 presents calculations by SWARA method. The percentages from Table 4 were applied to the final sub-criteria (Table 6, 1st and 6th column).

**Table 7. The criteria weight aggregation during I-II assessment.**

| | Arithmetic mean after normalization MaxMin c1/c2/c3/c4/c5/c6 criteria groups | | Arithmetic mean after normalization MaxMin of each criterion group | | | | | | | |
| | | | c1 criteria group | | c2 criteria group | | c4 criteria group | | c3/c5/c6 criteria groups | |
| | I | II | I | II | I | II | I | II | I | II |
|---|---|---|---|---|---|---|---|---|---|---|
| 1 | 0.4712 | 0.7109 | 0.5551 | 0.8278 | 0.3404 | 1.0000 | 0.4943 | 0.4943 | 0.4819 | 0.4819 |
| 2 | 0.5179 | 0.7959 | 0.5981 | 0.7992 | 0.4916 | 0.8320 | 0.4210 | 0.4210 | 0.3871 | 0.8579 |
| 3 | 0.3795 | 0.6338 | 0.4249 | 0.8410 | 0.0840 | 0.5053 | 0.3971 | 0.3971 | 0.4899 | 0.3951 |
| 4 | 0.5312 | 0.8104 | 0.5407 | 0.5605 | 0.4702 | 0.8320 | 0.7914 | 0.7914 | 0.3871 | 1.0000 |
| 5 | 0.5260 | 0.6288 | 0.3729 | 0.7281 | 0.5542 | 0.4947 | 0.6057 | 0.6057 | 0.4899 | 0.6794 |
| 6 | 0.3325 | 0.4073 | 0.2681 | 0.4962 | 0.2489 | 0.3573 | 0.4676 | 0.4676 | 0.4819 | 0.4819 |
| 7 | 0.5099 | 0.6382 | 0.4405 | 0.5035 | 0.5542 | 0.7374 | 0.6057 | 0.6057 | 0.3951 | 0.6794 |
| 8 | 0.3169 | 0.4590 | 0.4303 | 0.7250 | 0.1435 | 0.2733 | 0.3000 | 0.3000 | 0.4154 | 0.4627 |
| 9 | 0.4583 | 0.7841 | 0.4460 | 0.6109 | 0.4107 | 0.8320 | 0.6438 | 0.6438 | 0.3871 | 0.8579 |
| 10 | 0.3753 | 0.6468 | 0.3840 | 0.4033 | 0.0595 | 0.5480 | 0.6095 | 0.6095 | 0.3951 | 1.0000 |
| 11 | 0.4647 | 0.6341 | 0.3543 | 0.3755 | 0.2809 | 0.6320 | 0.6057 | 0.6057 | 0.5846 | 1.0000 |
| 12 | 0.4644 | 0.7443 | 0.2956 | 0.5182 | 0.3649 | 0.8320 | 0.6057 | 0.6057 | 0.4899 | 0.8579 |
| 13 | 0.5605 | 0.6102 | 0.4492 | 0.6121 | 0.3924 | 0.4520 | 0.5733 | 0.5733 | 0.9052 | 0.8579 |
| 14 | 0.4209 | 0.5159 | 0.3633 | 0.3246 | 0.3329 | 0.4520 | 0.5790 | 0.5790 | 0.6129 | 0.8579 |
| 15 | 0.4667 | 0.7617 | 0.4704 | 0.8409 | 0.2809 | 0.7267 | 0.6095 | 0.6095 | 0.5846 | 0.6794 |
| 16 | 0.3609 | 0.4660 | 0.2251 | 0.3611 | 0.3329 | 0.2733 | 0.2228 | 0.2228 | 0.5846 | 1.0000 |
| 17 | 0.3440 | 0.6912 | 0.5300 | 0.4850 | 0.1435 | 0.8320 | 0.5086 | 0.5086 | 0.1976 | 1.0000 |
| 18 | 0.5866 | 0.5973 | 0.5651 | 0.5738 | 0.4702 | 0.4947 | 0.7476 | 0.7476 | 0.7077 | 0.6603 |
| 19 | 0.2935 | 0.4473 | 0.5307 | 0.7567 | 0.0595 | 0.0840 | 0.3200 | 0.3200 | 0.0948 | 0.4819 |
| 20 | 0.5118 | 0.8179 | 0.4127 | 0.7549 | 0.3618 | 0.8320 | 0.5086 | 0.5086 | 0.4899 | 0.8579 |
| 21 | 1.0000 | 1.0000 | | | | | | | | |

Abbreviations: I- First assessment, II- Second assessment; 21 row stands for healthy person.

**Aggregation method for the decision support system.** The manuscript proposes an MCDM methodology for addressing real-life rehabilitation problems. This methodology aids rehabilitation specialists in determining the success of rehabilitation in individual cases. Patients are categorized into classes based on the success of their recovery employing both the heuristic method and the K-Means clustering algorithm after ranking with the weighted aggregation method [43]. This classical weighted aggregation is commonly referred as the simple additive weighting method. The weighted aggregation method (WAM) is implemented through the following steps:

- Firstly, we constructed a situation vector for each patient and include it to the table. In this table, values $x_{ij}$ are presented using the initial information, which includes patient evaluations in columns $j$ before and after rehabilitation, utilizing a set of the corresponding criteria in rows $i$.

- Secondly, after the construction and presentation of patient situation vectors in the tables, we normalize the patient data using the compromise normalization equation [44]: $r_{ij} = \frac{x_{ij} - \min_i x_{ij}}{\max_i x_{ij} - \min_i x_{ij}}$ (For maximized criterion),

$$r_{ij} = \frac{\max_i x_{ij} - x_{ij}}{\max_i x_{ij} - \min_i x_{ij}} \text{ (For minimized criterion)}.$$

**Table 8. Distribution of patients in classes according to the opinion of experts.**

| Patients | I assessment | 1 Expert | 2 Expert | 3 Expert | II assessment | 1 Expert | 2 Expert | 3 Expert |
|---|---|---|---|---|---|---|---|---|
| | | | | Assigning patients to a class according to experts' opinion (class) | | | | |
| 1 | II class | II | III | II | IV class | III | IV | IV |
| 2 | III class | III | III | III | IV class | IV | IV | IV |
| 3 | II class | II | II | II | III class | III | III | III |
| 4 | III class | III | III | III | IV class | IV | IV | IV |
| 5 | III class | III | III | III | III class | III | III | III |
| 6 | II class | II | II | II | II class | II | II | III |
| 7 | III class | III | III | IV | III class | III | III | III |
| 8 | II class | II | II | II | II class | II | II | II |
| 9 | II class | II | II | III | IV class | IV | IV | IV |
| 10 | II class | II | II | II | III class | III | III | III |
| 11 | II class | II | III | II | III class | III | III | III |
| 12 | II class | II | II | II | IV class | IV | IV | IV |
| 13 | III class | III | III | III | III class | III | III | III |
| 14 | II class | II | II | II | III class | III | III | III |
| 15 | II class | II | III | II | IV class | IV | IV | IV |
| 16 | II class | II | II | II | II class | III | III | II |
| 17 | II class | II | II | II | IV class | III | IV | IV |
| 18 | III class | III | IV | III | III class | III | III | III |
| 19 | II class | III | III | III | II class | III | III | III |
| 20 | III class | III | III | III | IV class | IV | IV | IV |

- Thirdly, the normalized data is processed to achieve the classification of patients based on rehabilitation success and facilitate the final decision. These calculations are executed using the WAM formula:

$$\text{WAM} = \sum_{j=1}^{n} \left( w_j \, r_{ij} \right),$$

where $w_j = (w_1, w_2 \ldots w_n)$ is vector of criteria weights

## Results

The mean age of the 20 patients in the study was 64.3 ± 10.5 years (min 55, max 88 years), and 12 (60.0%) were women. 55% of patients had higher education, 30% secondary, and 15% had primary. Additionally, 80% had working status. The time since surgery was measured from the onset of the first pain symptoms or shoulder dysfunction. The dominant arm affected 75% of patients, specifically the right hand. The damaged muscles were identified as follows: m. supraspinatus for 45% of patients, m. infraspinatus 8%, m. subscapularis 2%. combined (m. supraspinatus and m. infraspinatus) for 30% of patient, more than three demerged muscles for 15% of patients. The main reasons for RC tears were: (1) degenerative changes in RC muscles 45% of patients (heavy weightlifting (20%), constant repetitive shoulder loading (50%), sudden movement over the shoulder (15%), falls (from cycling, walking, running, rushing) (15%)), (2) shoulder damage during trauma 35% of patients and (3) unknown causes in 20% of patients.

The application of the methodological steps including normalization and weighting, in the context of rehabilitation outcomes, is illustrated through the creation of a decision-making

**Table 9. Classification of patients into classes according to rehabilitation success.**

| Classification of patients into classes | | I class | II class | III class | IV class | V class |
|---|---|---|---|---|---|---|
| | | 0.0–0.29 | 0.3–0.49 | 0.5–0.69 | 0.7–0.89 | 0.9–0.99 |
| Patient's distribution between classes n (%) | I assessment | 0 (0%) | 13 (65%) | 7 (35%) | 0 (0%) | 0 (0%) |
| | II assessment | 0 (0%) | 4 (20%) | 8 (40%) | 8 (40%) | 0 (0%) |

matrix for the primary rehabilitation protocol. This matrix provides the final value for each patient at the beginning and end of rehabilitation (S1 Table). S2 Table represents the normalized weighted decision-making matrix of 20 patients during the initial (I) and (II) assessment of rehabilitation outcomes. To assess the changes in patients' recovery results and compare them with the condition of a healthy person, we introduced the 21st patient to the matrix and representing a healthy individual.

Table 7 illustrates the aggregation of common criteria weights for 20 patients during the I-II assessment, providing a detailed analysis of the weights' aggregation for each group of criteria (arithmetic mean after normalization MaxMin). Additionally, Table 7 includes the aggregation of criteria weights in the 21st position.

After conducting a detailed analysis of the weighted aggregation results for individual criteria of each patient group before and after rehabilitation, experts were tasked with evaluating the rehabilitation outcome of each patient by categorizing them into classes. The assignment of patients into classes, based on expert opinions, did not differ significantly from our employed patient classification method (Table 8). Following discussion with the experts, a consensus was reached that our i classification of patients into 5 classes aligns with the expectations and requirements of rehabilitation specialists, considering individual evaluation criteria.

The final result of our decision support methodology reveals that at the outset of rehabilitation 90% of patients exhibited severe and moderate impairment, experiencing functional problems in daily life activities and were categorized into II-III class (Table 9). The effectiveness of rehabilitation is indisputable, with a significant improvement in shoulder motor function, a reduction in pain and diminished disability noted at the end of rehabilitation. 80% of patients displayed moderate and mild impairment, thus being assigned to III and IV classes (Table 9).

After the expert assessment, the evaluation and verification of the class boundary values were caried out using the K-Means clustering method. K-Means clustering is a classic unsupervised learning algorithm applicable to grouping diverse data, such as patient data, based on their health status [45]. The K-Means calculations are conducted as follows:

1. The number of clusters $K$ (in our case is five);

2. Cluster centroids $c_i$ are initialized by k-means++ algorithm [46];

3. $D = d(x_i, c_i)$—the Euclidean distance is calculated between the cluster centroids and each data point;

4. According to the calculated D, all data points are assigned to the nearest centroid;

5. All cluster centroid positions $c_i$ are recalculated by computing the mean *of currently assigned data points*:

$$c_i = 1/|C_i| \cdot \sum_{x \in C_i} x_i$$

**Table 10. Evaluation of the patient class boundary values based on experts and the K-Means clustering method.**

| Number of classes | Boundaries of classes by experts | Boundaries of classes by K-Means for corrected interval values |
|---|---|---|
| 1 | [0.0100–0.3168] | [0.0100–0.2934] |
| 2 | [0.3169–0.491] | [0.2935–0.491] |
| 3 | [0.492–0.668] | [0.492–0.668] |
| 4 | [0.6692–0.8179] | [0.669–0.8179] |
| 5 | [0.818–1] | [0.818–1] |

1. The cycle from the third step is repeated until the position of the cluster centroids no longer changes.

The larger the sample of different patients, the more rehabilitation values we have, allowing for a precise determination of the cut-off values for the classes. Therefore, we combine expert assessment before and after rehabilitation. In each case, we have 21 rehabilitation assessments of patients. For the 5 classes of patients, the minimum and the maximum values (intervals) according to experts are presented in Table 9 in the 2nd column. There is one outlier in the expert assessment (value **0.2935)** that is rejected.

After grouping with the K-Means clustering algorithm of the pre-and post-rehabilitation patient values acquired after MCDM processing, we obtain intervals for the fifth classes (Table 9, column 3). The range of values for patients before and after rehabilitation after WAM calculations was [0.2935–1]. We included an additional min value of 0.01 in this range of values. We assumed that if we have a maximum possible value of 1, we should also have a minimum possible value of about 0). Finally, using K-Means clustering for the corrected interval of values (Table 9, column 4), we got rehabilitation ranges of values for the five classes that coincide with the expert evaluation, except for class 1.

In order to avoid gaps between class intervals, we adjust the min and the max values of the intervals as follows:

1. From each min value of the following class interval, we subtract the max value of the preceding class interval and divide the difference in half;

2. Further, we add the obtained value to the max value of the lower class and subtract it from the min value of the higher class.

Looking at the results in Table 10, we see that the heuristic evaluation method based on expert opinion and the standard qualitative evaluation methods shows that the intervals for all five classes are similar and do not significantly differ. It is important to note that the applicability of MCDM methods in the field of rehabilitation is acceptable and reliable, and the use of a decision support system would greatly facilitate the work of rehabilitation professionals.

## Discussion

The proposed methodology was applied in a case study in the rehabilitation protocol selection after RC repair. The novelty of our research is the development and testing of a new decision support system that combines many factors into a single valuation and allows each patient to assess his capabilities separately according to individual limitations of physical, functional and psycho-emotional functions. The aggregation of the multiple criteria and detailed analysis of factors influencing recovery help to determine and decide on a successful rehabilitation protocol for patients after RC repair.

The reader may naturally wonder why current traditional rehabilitation methods are not suitable. However, they are indeed appropriate. The application of traditional assessment methods used in rehabilitation is old and reliable and supported by numerous studies demonstrating their effectiveness. However, the introduction of innovations and the search for more accurate methods are necessary to standardize the assessment strategy among specialists, establish a common language and terminology, and create a system/algorithm based on mathematical calculations to eliminate the subjective (assessor's) approach. Therefore, our case study experts, who have many years of experience in rehabilitating such types of patients, selected the factors most influencing the recovery of functions, evaluated the patients and repeated the evaluation after rehabilitation. The selection of criteria was based on the results of recent scientific research and the long-term rehabilitation experience of experts in restoring impaired biopsychosocial functions of patients after RC tears. During the expert's discussion, the criteria and sub-criteria were defined, and after being ranked by importance, their weighting coefficients were calculated, and data processing was performed using multiple-criteria decision-making methods. MCDM methods calculated the results according to the applied algorithms, and the experts were asked again to check and review whether the mathematical methods correctly presented the data and grouped the patients into classes. Due to the detailed weight aggregation results (Tables 6 and 7) of individual criteria for each patient group, it is possible to individually assess which function has the most significant recovery during rehabilitation, which physical or functional problems still lead to impairment and disability, or have led to a worsening of the situation. Based on the received data, the specialist can adjust and analyze the individual rehabilitation protocol: how the applied measures influence the rehabilitation results, what primary factors (age, injury or damaged muscle) determine the choice of the protocol, what factors had the greatest therapeutic effect, what is the patients' physical tolerance and functional recovery. It was also assessed whether the groups formed by experts and those proposed by statistical methods differed significantly. Our study showed that a very similar result was obtained after conducting a heuristic evaluation and applying standard methods. There was not much difference in the distribution of patients into classes, and the class boundaries were similar, which shows us that the MCDM methods are accurate and close and can even be said to correspond to the experience of rehabilitation specialists. This allows us to assume that MCDM methods are sensitive and flexible, allowing for variation according to different pathologies and individualized interventions and programs. However, the most important thing is that this method allows you to simultaneously see the general picture of the patient and accurately record the changes, as well as monitor the progress or analyze in detail which factor at which time influenced the healing. And thus, it is possible to detail the influencing factors and adjust and apply the rehabilitation program to each specific situation. Evaluation of rehabilitation results and the general physical and functional condition of patients before and after the rehabilitation program was carried out using a multi-criteria support system. The values of each patient's criteria influencing recovery and overall progress were analyzed, the expertise performed, and the results obtained were calibrated with the data of statistical methods, the obtained results of which showed only insignificant differences between assigning criteria weights and classifying patients into classes. This shows the possibility of carrying out a detailed analysis of the criteria/factors influencing the situation of each patient individually, taking into account his condition, rehabilitation time or healing process.

The novelty of MCDM methods lies in their continuous evolution and development, with new approaches and techniques being proposed and improved to better handle increasingly complex decision-making tasks too. In rehabilitation, this allows for a wide range of choices and a flexible decision-making algorithm when analyzing or correcting the rehabilitation protocol.

In our results, we only see the advantages of the methods we used, so we recommend using decision-making systems in rehabilitation in different areas so that we can see the most suitable methods, algorithms, and protocols. Also, in order to determine the limitations of decision-making systems in rehabilitation, we must monitor and continuously analyze the use of these systems in practice for many years.

## Conclusions

After the case study using the MCDM methodology, we can see that it perfectly contributes to the decision support system, thus improving the efficiency of parameters selection and allowing young. inexperienced specialists to manage the rehabilitation process more easily. Our results confirmed that the decision support system could be perfectly applied in rehabilitation when selecting protocols, analyzing different factors influencing rehabilitation outcomes, and predicting the patient's recovery limits.

In summary, firstly, the multi-criteria support system allows us to evaluate the effectiveness of rehabilitation and to monitor changes in the general physical and functional condition of patients during rehabilitation. Secondly, according to the result of the created support system, it is possible to assess the condition or the course of recovery according to the criteria values of individual patients. Finally, the similar evaluation results obtained between expert opinion and statistical methods show an excellent opportunity to analyze each criterion separately, detailing and individualizing according to different pathology, health status or recovery perspective.

## Supporting information

**S1 Table. Rehabilitation protocol decision-making matrix (I and II assessments, points).**
(PDF)

**S2 Table. Weighted decision-making matrix during I-II assessments.**
(PDF)

**S1 Data. A minimal data set file is also attached: Minimal data set.**
(XLSX)

## Author Contributions

**Conceptualization:** Aušra Adomavičienė, Kristina Daunoravičienė, Girūta Kazakevičiūtė-Januškevičienė, Romualdas Baušys.

**Data curation:** Aušra Adomavičienė.

**Formal analysis:** Aušra Adomavičienė, Girūta Kazakevičiūtė-Januškevičienė, Romualdas Baušys.

**Investigation:** Aušra Adomavičienė.

**Methodology:** Aušra Adomavičienė, Romualdas Baušys.

**Project administration:** Aušra Adomavičienė.

**Resources:** Aušra Adomavičienė.

**Supervision:** Aušra Adomavičienė.

**Validation:** Aušra Adomavičienė, Kristina Daunoravičienė, Girūta Kazakevičiūtė-Januškevičienė, Romualdas Baušys.

**Visualization:** Kristina Daunoravičienė.

**Writing – original draft:** Aušra Adomavičienė, Kristina Daunoravičienė, Girūta Kazake-
vičiūtė-Januškevičienė, Romualdas Baušys.

**Writing – review & editing:** Kristina Daunoravičienė.

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
