## [Decision Letter · Decision Letter 0]

10 Nov 2023

PONE-D-23-31967Functional recovery prediction during rehabilitation after cuff tears by decision support

systemPLOS ONE

Dear Dr. Daunoraviciene,

Thank you for submitting your manuscript to PLOS ONE. After careful consideration, we feel that it has merit but does not fully meet PLOS ONE’s publication criteria as it currently stands. Therefore, we invite you to submit a revised version of the manuscript that addresses the points raised during the review process.

We look forward to receiving your revised manuscript.

Kind regards,

Shabnam ShahAli, Ph.D.

Academic Editor

PLOS ONE

5. We notice that your supplementary [figures/tables] are included in the manuscript file. Please remove them and upload them with the file type 'Supporting Information'. Please ensure that each Supporting Information file has a legend listed in the manuscript after the references list.

Reviewers' comments:

Reviewer's Responses to Questions

**Comments to the Author**

1. Is the manuscript technically sound, and do the data support the conclusions?

Reviewer #1: Yes

Reviewer #2: Yes

Reviewer #3: Yes

2. Has the statistical analysis been performed appropriately and rigorously? 

Reviewer #1: Yes

Reviewer #2: Yes

Reviewer #3: Yes

3. Have the authors made all data underlying the findings in their manuscript fully available?

Reviewer #1: Yes

Reviewer #2: Yes

Reviewer #3: Yes

4. Is the manuscript presented in an intelligible fashion and written in standard English?

Reviewer #1: No

Reviewer #2: Yes

Reviewer #3: Yes

5. Review Comments to the Author

Reviewer #1: Dear Dr Adomavičienė,

Please note and correct the following:

1. There are some typing errors in the manuscript. Please review and adjust accordingly. line 150 (treatmens, treatments), line 290 (“each criteria” should be changed to “each criterion”), line 265 (injury related characteristics, injury-related characteristics) …….

2. The Step-wise Weight Assessment Ratio (SWARA) should be used the first time it appears in the manuscript on page 5 and line 119.

3. In the participant section, please clarify where the participants were sampled, instead of saying “our outpatient department” in line 224.

4. In the exclusion criteria section, why did you exclude patients over 8 weeks after surgery?

5. In Table 2: why does the table footer contain “NA” while this abbreviation does not appear in the table body?

6. In the references section: The names of the journals of references 36, 37, 42, and 44 should be written in abbreviated form.

Reviewer #2: it is acceptable after minor revision.please edit your introduction.....................................................................................................................................................

Reviewer #3: This article investigates Functional recovery prediction during rehabilitation after cuff tears by decision support system. It is an interesting and new topic. The statistical analysis has been performed appropriately and the manuscript written in standard English.

The comments below may help you rewrite the article.

Abstract:

- The method section does not explain the type of study and exact methodology, (including the number of patients, rehabilitation treatment, etc.). It seems better to add these items.

- Nothing has been written about MCDM in the results section.

Main text

In the participants’ section:

- where were the rehabilitation treatments done? The name of the center and...

- Why do you insert 24 patients? How was the sample size calculated?

- In line 219, you wrote 6-7 weeks after surgery; however, in line 225, you wrote 5-6 weeks after surgery...it is not clear to the reader….

- Please insert the registration number and URL.

- Please insert the ethical code.

In the follow-up routine section:

- What was the home exercise program? Was it done daily? How were the home exercises checked?

Results section:

- Is it not necessary to write the demographic information in a table? Do demographic characteristics not affect treatment response?

- Maybe it's better to put a study flow diagram.

6. PLOS authors have the option to publish the peer review history of their article (what does this mean?). If published, this will include your full peer review and any attached files.

Reviewer #1: **Yes: **Mahnaz Tavahomi

Reviewer #2: **Yes: **Soheil Mansour Sohani

Reviewer #3: No

---

## [Author Response · Author response to Decision Letter 0]

13 Dec 2023

Response to Academic Editor

We paid attention to fulfil all comments of the Academic Editor, which have greatly helped us to improve our manuscript based on journal requirements. We hope that we have improved our article enough to make it suitable for scientific publication in the journal “PLOS ONE”. Please find enclosed the revised manuscript “Functional recovery prediction during rehabilitation after rotator cuff tears by decision support system”.

Thank you very much.

Sincerely,

Authors

Point 1: Please ensure that your manuscript meets PLOS ONE's style requirements, including those for file naming. The PLOS ONE style templates can be found at 

Response 1: the manuscript was carefully reviewed and corrected according to the requirements of the journal. 

Point 2: Note from Emily Chenette, Editor in Chief of PLOS ONE, and Iain Hrynaszkiewicz, Director of Open Research Solutions at PLOS: Did you know that depositing data in a repository is associated with up to a 25% citation advantage (https://doi.org/10.1371/journal.pone.0230416)? If you’ve not already done so, consider depositing your raw data in a repository to ensure your work is read, appreciated and cited by the largest possible audience. You’ll also earn an Accessible Data icon on your published paper if you deposit your data in any participating repository (https://plos.org/open-science/open-data/#accessible-data).

Response 2: to increase data availability, a minimum data set was created and attached to the Supporting Information files.

Point 3: Please amend either the title on the online submission form (via Edit Submission) or the title in the manuscript so that they are identical.

Response 3: The title on the online submission form was updated and currently they are identical with title in the manuscript.

Point 4: Please include your full ethics statement in the ‘Methods’ section of your manuscript file. In your statement, please include the full name of the IRB or ethics committee who approved or waived your study, as well as whether or not you obtained informed written or verbal consent. If consent was waived for your study, please include this information in your statement as well. 

Response 4: The required information was added to ‘Methods’ section. 

The data collection followed Ethical Protocol No KFVDSGS2021 provided by the Vilnius Regional Biomedical Research Ethics Committee (No. 2021/5-1349822), and informed written consent was obtained. 

Point 5: We notice that your supplementary [figures/tables] are included in the manuscript file. Please remove them and upload them with the file type 'Supporting Information'. Please ensure that each Supporting Information file has a legend listed in the manuscript after the references list.

Response 5 Appendix tables have been separated from the publication and provided as Supporting Information files in pdf format S1 table and S2 table. Supporting Information is detailed at the end of the publication and references to tables in the text have been updated.

Point 6: Please include captions for your Supporting Information files at the end of your manuscript, and update any in-text citations to match accordingly. Please see our Supporting Information guidelines for more information: http://journals.plos.org/plosone/s/supporting-information. 

Response 6 Supporting Information is detailed at the end of the publication and references to tables in the text have been updated.

Point 7. Please review your reference list to ensure that it is complete and correct. If you have cited papers that have been retracted, please include the rationale for doing so in the manuscript text, or remove these references and replace them with relevant current references. Any changes to the reference list should be mentioned in the rebuttal letter that accompanies your revised manuscript. If you need to cite a retracted article, indicate the article’s retracted status in the References list and also include a citation and full reference for the retraction notice.

Response 7 We have reviewed and updated the reference list.

Response to Reviewer 1 Comments

Dear Reviewer,

We are grateful for your valuable comments and comments, which have greatly helped us to improve our manuscript. We hope that we have responded efficiently to all comments and all the questions, and also improved our article enough to make it suitable for scientific publication in the journal PLOS ONE. 

Thank you very much.

Sincerely,

Authors

Point 1: There are some typing errors in the manuscript. Please review and adjust accordingly. line 150 (treatmens, treatments), line 290 (“each criteria” should be changed to “each criterion”), line 265 (injury related characteristics, injury-related characteristics).

Response 1: Thank you for comments. The manuscript was corrected. 

Point 2: the Step-wise Weight Assessment Ratio (SWARA) should be used the first time it appears in the manuscript on page 5 and line 119.

Response 2: Thanks for the note. The full name was given “The Step-wise Weight Assessment Ratio – SWARA”.

Point 3: In the participant section, please clarify where the participants were sampled, instead of saying “our outpatient department” in line 224.

Response 3: The information in the ‘Participants’ subsection was clarified. The prospective cohort clinical study included 20 patients after RC surgery, undergoing outpatient rehabilitation at Vilnius University Hospital's Santara Clinic Rehabilitation Center.

Point 4: In the exclusion criteria section, why did you exclude patients over 8 weeks after surgery?

Response 4: We do not include patients over > 8 weeks after surgery, because we wanted to ensure the same recovery conditions for patients during the study. Typically, after RC surgery, patients wear an abduction brace for 6 weeks until the tendon heals, and primary rehabilitation begins at week 7. if patients come to our center after 8-10 weeks, it means that they are coming for secondary rehabilitation and are well advanced in terms of recovery.

Point 5: In Table 2: why does the table footer contain “NA” while this abbreviation does not appear in the table body?

Response 5: Thanks for your note. “NA” abbreviation was deleted, it was a mistake. 

Point 6: In the references section: The names of the journals of references 36, 37, 42, and 44 should be written in abbreviated form.

Response 6: Thanks. The names of the journals of references 36, 37, 42, and 44 were corrected in the reference list. 

Response to Reviewer 2 Comments

Dear Reviewer,

We are grateful for your time. We hope that we have responded efficiently to your comment and improved our article enough to make it suitable for scientific publication in the journal PLOS ONE. 

Thank you very much.

Sincerely,

Authors

Point 1: it is acceptable after minor revision. please edit your introduction

Respond 1: Thanks for your comment. We tried to do as much as we could to improve our manuscript language. 

Response to Reviewer 3 Comments

Dear Reviewer,

We appreciate your time and valuable comments. This helped us a lot to improve the manuscript. We hope that we have effectively responded to all comments and questions, and improved our article sufficiently to be eligible for scientific publication in PLOS ONE.

Thank you very much.

Sincerely,

Authors

Point 1: Abstract:

- The method section does not explain the type of study and exact methodology, (including the number of patients, rehabilitation treatment, etc.). It seems better to add these items.

Response 1: In the method section was explained the type of study and exacted methodology. Methods: For development of the decision support system constructed applying the MCDM framework 20 patients after RC operation undergoing the outpatient rehabilitation were enrolled in the prospective cohort clinical trial. 

Point 2: Nothing has been written about MCDM in the results section.

Response 2: Thank you for a very important and to-the-point comment. We have added information in the Results section.

The results of developing a decision support methodology for predicting functional recovery outcomes at early rehabilitation after RC repair, show that applying the MCDM framework (SWARA method) sensitively assesses the different criteria and determines the corresponding criteria weights that were similar to criteria weights assessed subjectively by rehabilitation experts. After criteria weight aggregation patients were divided into 5 classes. The assignment of patients into the classes, according to the heuristic evaluation method based on expert opinion and the standard qualitative evaluation methods shows that the intervals for all five classes are similar and do not significantly differ.

Point 3: Main text

In the participants’ section:

- where were the rehabilitation treatments done? The name of the center and...

Response 3: The information in the ‘Participants’ subsection was clarified. The prospective cohort clinical study included 20 patients after RC surgery, undergoing outpatient rehabilitation at Vilnius University Hospital's Santara Clinic Rehabilitation Center.

Point 4: - Why do you insert 24 patients? How was the sample size calculated?

Response 4: We would like to clarify that our study was conducted from 21.06.2021 to 20.12.2022 (It is presented in subsection Participants. The recruitment period for this study from 21.06.2021 to 20.12.2022.) Longitudinal monitoring of patients' functional recovery was carried out for 6 months, i.e., follow-ups are performed after 1, 3 and 6 months. Only 20 patients participated in all phases of the longitudinal study. We have provided only the first data of the primary rehabilitation recovery data (6-7 weeks after surgery) and for development of support - assistant system it is needing the same functional status of patients and, therefore, the same inclusion criteria are necessary. That’s is why only 20 patients' data are presented in the final data analysis for this publication. 

The samples size was preliminary defined due to the study objectives and the specifics of the MCDM method used. This type of research can be conducted for case analysis separately, so the sample size calculation in our study will not affect the significance of the results.

Point 5: - In line 219, you wrote 6-7 weeks after surgery; however, in line 225, you wrote 5-6 weeks after surgery...it is not clear to the reader….

Response 5: Thanks for your comment. It was a mistake, we corrected into “6-7 weeks after surgery”

Point 6: - Please insert the registration number and URL.

- Please insert the ethical code.

Response 6: The required information was added to ‘Methods’ section. 

The data collection followed Ethical Protocol No KFVDSGS2021 provided by the Vilnius Regional Biomedical Research Ethics Committee (No. 2021/5-1349822), and informed written consent was obtained. 

Point 7: In the follow-up routine section:

- What was the home exercise program? Was it done daily? How were the home exercises checked?

Response 7: During rehabilitation, patients are given an additional exercise program at home (a complex of exercises that is selected and composed by a physiotherapist for the patient according to his functional condition, physical capacity, and load tolerance as an additional means of increasing the load). At the end of the rehabilitation program, the patients are given an exercise program at home for 2 weeks, then the home program is adjusted during the patient's visits to the physiotherapist after 2 weeks 1,3, or 6 months.

Point 8: Results section:

- Is it not necessary to write the demographic information in a table? Do demographic characteristics not affect treatment response?

- Maybe it's better to put a study flow diagram.

Response 8: Thanks for the comment, we have taken it into consideration. Taking into account the fact that in our work there are already so many tables that it can be quite confusing for the reader, we have presented the obscured patient data in text. Since we already have one flow chart in Fig 1, we consulted and decided not to add a new one, but simply explain it in the text.

The mean age of the 20 patients in the study was 64.3 ± 10.5 years (min 55, max 88 years), and 12 (60.0%) were women. 55% of patients had higher education, 30% secondary, and primary 15%. 80% had working status. The mean elapsed time since the index surgery was: 1-2 weeks 7(35%), 3-4 weeks 4 (20%), and >5-6 weeks 9 (45%). The dominant arm affected 75% of patients and affected the right hand. Damaged m. supraspinatus for 45% of patients, m. infraspinatus 8%, m. subscapularis 2% combined (m. supraspinatus and m. infraspinatus) for 30% of patients and more than three demerged muscles for 15% of patients. The main reasons for RC tears were related to degenerative RC muscle changes 45%: heavy weightlifting (20%), constant repetitive load on the shoulder (50%), falls (from cycling, walking, running, rushing) (15%), making a sudden movement over the shoulder (15%), damaged shoulder during trauma 35% and unknown causes for 20% patients.

---

## [Decision Letter · Decision Letter 1]

26 Dec 2023

Functional recovery prediction during rehabilitation after rotator cuff tears by decision support system

PONE-D-23-31967R1

Dear Dr. Daunoraviciene,

We’re pleased to inform you that your manuscript has been judged scientifically suitable for publication and will be formally accepted for publication once it meets all outstanding technical requirements.

Kind regards,

Shabnam ShahAli, Ph.D.

Academic Editor

PLOS ONE

Reviewers' comments:

Reviewer's Responses to Questions

**Comments to the Author**

1. If the authors have adequately addressed your comments raised in a previous round of review and you feel that this manuscript is now acceptable for publication, you may indicate that here to bypass the “Comments to the Author” section, enter your conflict of interest statement in the “Confidential to Editor” section, and submit your "Accept" recommendation.

Reviewer #1: All comments have been addressed

Reviewer #2: All comments have been addressed

2. Is the manuscript technically sound, and do the data support the conclusions?

Reviewer #1: Yes

Reviewer #2: Yes

3. Has the statistical analysis been performed appropriately and rigorously? 

Reviewer #1: Yes

Reviewer #2: Yes

4. Have the authors made all data underlying the findings in their manuscript fully available?

Reviewer #1: Yes

Reviewer #2: Yes

5. Is the manuscript presented in an intelligible fashion and written in standard English?

Reviewer #1: Yes

Reviewer #2: Yes

6. Review Comments to the Author

Reviewer #1: All my recommendations have been made carefully. However, it would be better to mention why not include patients over 8 weeks after surgery in the exclusion criteria section in brief (comment 4).

Reviewer #2: Congratulations

Your manuscript is publishable........................................................................................................................

7. PLOS authors have the option to publish the peer review history of their article (what does this mean?). If published, this will include your full peer review and any attached files.

Reviewer #1: No

Reviewer #2: **Yes: **Soheil Mansour Sohani

---

## [Editor Report · Acceptance letter]

11 Jan 2024

PONE-D-23-31967R1 

PLOS ONE

Dear Dr. Daunoravičienė, 

I'm pleased to inform you that your manuscript has been deemed suitable for publication in PLOS ONE. Congratulations! Your manuscript is now being handed over to our production team.

Kind regards, 

on behalf of

Dr. Shabnam ShahAli 

Academic Editor

PLOS ONE